Geographic disparities and predictors of suicide mortality risk in Florida: spatial scan statistics and negative binomial modeling

Onyuth Howard
Day Corey
Deb Nath Nirmalendu
Odoi Agricola aodoi@utk.edu
Department of Biomedical and Diagnostic Sciences, University of Tennessee , Knoxville , TN , United States of America
Buyske Steven
Electronic publication date: 2025 Nov 4
Publication date: 2025
Volume: 13
Electronic Location ID: e20075
Received 2024 Oct 16; Accepted 2025 Aug 22
Copyright: ©2025 Onyuth et al.
Copyright year: 2025
Copyright holder: Onyuth et al.
License: This is an open access article distributed under the terms of the Creative Commons Attribution License, which permits unrestricted use, distribution, reproduction and adaptation in any medium and for any purpose provided that it is properly attributed. For attribution, the original author(s), title, publication source (PeerJ) and either DOI or URL of the article must be cited.
License URL: https://creativecommons.org/licenses/by/4.0/

Keywords: Geographic disparities, Spatial empirical Bayesian smoothing, Suicide, Suicide mortality risk, Tango’s flexible spatial scan statistics, Geographic information systems, GIS

Funding: The authors received no funding for this work.

==============================
Background

Florida ranks 17th in suicide mortality risk in the United States, with 14.8 suicide deaths per 100,000 persons. Geographic disparities in suicide mortality across the US are well-documented and are partly attributed to the uneven distribution of risk factors. However, limited data exist on such disparities and associated predictors within Florida, despite their importance for guiding targeted prevention efforts. This study aimed to investigate county-level geographic disparities in suicide mortality risk in Florida and identify predictors of these disparities.

Methods

This retrospective ecological study used data from the Florida Department of Health. County-level age-adjusted suicide mortality risks and spatial empirical Bayesian-smoothed risks were calculated for three time periods: 2011–2013, 2014–2016, and 2017–2019. Tango’s spatial scan statistics were applied to identify high-risk clusters. A negative binomial regression model was used to examine county-level predictors of suicide mortality risk for the 2017–2019 period.

Results

Statewide age-adjusted suicide mortality risk increased from 22.6 to 24.3 per 100,000 persons over the study period. Counties in the northwest, northeast, southwest, and parts of central Florida consistently exhibited high mortality risks. Suicide mortality risk was significantly higher in counties with larger proportions of residents aged 45–64 years and ≥65 years, those reporting excessive drinking, frequent mental distress, or veteran status.

Conclusion

This study identified geographic disparities and key predictors of suicide mortality risk across Florida counties. These findings can inform policymakers, healthcare providers, and community organizations in designing and implementing targeted suicide prevention programs tailored to the specific needs of high-risk communities.

Introduction

Suicide, defined as the act of intentionally causing one’s own death (Nock et al., 2008; Moutier, 2024), has increasingly become a major public health concern across the globe (Facioli et al., 2015). Approximately 800,000 people worldwide die of suicide annually (Garnett & Curtin, 2023) implying that, on average, one person dies from suicide every 40 s (Martinez-Ales et al., 2020). In 2021, suicide was the 18th leading cause of death globally and the 11th in the United States (US) (Saman et al., 2012; Hisham et al., 2023). In the US, as many as 32,000 people take their own lives every year, which amounts to about 132 suicides daily or one suicide every 11 min (Centers for Disease Control and Prevention , 2024).

The annual economic burden attributable to suicide and non-fatal self-harm in the US is estimated at $510 billion (Peterson, Haileyesus & Stone, 2024). Most of this is due to the high cost of life-years-lost ($484 billion) followed by medical spending ($13 billion), reduced quality of life ($10 billion), and work loss due to non-fatal injuries ($3 billion) (Hansen & Gryglewicz, 2016; Peterson, Haileyesus & Stone, 2024). The annual cost of suicides in Florida is estimated at $2.84 billion (Hansen & Gryglewicz, 2016).

Despite the high economic and societal costs of suicide, very little is currently known about the geographic disparities and predictors of suicide mortality risks in Florida (Tøllefsen, Hem & Ekeberg, 2012; Peterson, Haileyesus & Stone, 2024). However, there is evidence of state-level geographic disparities in suicide mortality risks across the US. For example, Florida has the 17th highest suicide mortality risk in the US (Hansen & Gryglewicz, 2016; United Health Foundation, 2024) and suicide is the 12th leading cause of death in the state (Florida Department of Children and Families, 2024). Unfortunately, there is currently very sparse information on disparities in suicide mortality risks within the state and yet this information is essential for developing targeted interventions and guiding resource allocation for suicide prevention programs (Saman et al., 2012). Identification of geographic disparities may help identify regions with significantly high suicide mortality risks. This would help guide targeted control efforts to curb the problem and reduce/eliminate disparities (Saman et al., 2012). Therefore, the objectives of this study were to investigate county-level geographic disparities in suicide mortality risks in Florida and identify predictors of these disparities so as to guide targeting of control efforts to areas that are most in need.

Materials & Methods

Ethics review

This study was reviewed by the University of Tennessee Institutional Review Board (IRB) which determined that it was not human subjects research (IRB Number: UTK IRB-24-083540-XM) because the secondary data used in the study did not contain information on living individuals. Therefore, the board determined that IRB oversight was not required.

Study area

This study was conducted in Florida, a US state composed of 67 counties (Fig. 1). As of 2020, Florida’s population was estimated at 21.5 million, with the largest percentage residing in the southeast (32.1%), followed by southwest (20.8%), central (20.1%), northeast (11.0%), south (9.0%), and northwest (7.0%) regions (United States Census Bureau, 2020) (Fig. 1). Miami Dade, Broward, and Palm Beach, all located in the Southeast, are among the most populous counties in the state and collectively represent 65.3% of Florida’s population (Khan et al., 2022). Approximately 62.7% of the counties in Florida are considered mostly urban, 32.8% are mostly rural, and 4.5% are completely rural (UnitedStatesCensusBureau, 2010) (Fig. 1). This classification, based on the 2010 US Census Bureau County rurality levels, categorized counties as mostly urban if less than 50% of the population lived in rural areas, mostly rural if 50–99.9% of the population lived in rural area, and completely rural if 100% of the population lived in rural areas (UnitedStatesCensusBureau, 2010).

Figure 1 Maps of Florida showing county names, urban/rural classification of the counties, and regions.

Data sources

The secondary data used in the study were obtained from the Florida Department of Health (Florida Department of Health, 2024). The data were aggregated to county geographical scale and split into three 3-year time periods: 2011–2013, 2014–2016 and 2017–2019. Death records of all suicide deaths in Florida were used in the study. Deaths resulting from suicide were identified using the International Classification of Diseases 10th revision (ICD-10) External Causes of Death Codes X60-X84, and Y87.0 (Saman et al., 2012). The county-level population estimates were obtained from the 2012, 2015, and 2018 American Community Survey and were used as denominators for 2011–2013, 2014–2016, and 2017–2019 time periods, respectively (UnitedStates Census Bureau, 2023).

The potential predictor variables used in the analyses and data sources for each are listed in Table 1. The main data sources were the Florida Department of Health (Florida Department of Health, 2024), County Health Rankings and Road Map (Peterson et al. 2024; University of Wisconsin Population Health Institute, 2024), and the American Community Survey Veteran Status S2101 (Lavrakas, 2013). The 2023 Geographic Boundary Files for cartographic displays were obtained from the United States Census Bureau TIGER Geodatabase (United States Census Bureau, 2024).

Table 1 Data sources and variables used in the analysis investigating predictors of suicide mortality risks.

Data source	Variable	Level/Categories	Variable definition	
American Community Survey Veteran Status S2101	Veterans	Percentage	Percentage of Veterans	
Florida Department of Health	Age	Less than 18 years, 18 to 44 years, 45 to 64 years, and 65 years or over	Decedent’s age	
	Gender	Male and female	Sex of decedent	
	Marital status	Widowed, divorced, married and unknown marital status	Description of marital status of decedent	
	County of suicide	All the counties in Florida	Decedent’s county of residence	
	Suicide means	Self-poisoning, hanging, strangulation, suffocation, and drowning, firearm, and other means	International Classification of Diseases, Tenth Revision (ICD-10) code for underlying cause of death	
	Place of death	Inpatient, emergency room/outpatient, decedent’s home, hospice, and other places	Place of death	
	Education status	Less than high school education, high school, some college but no degree, associate degree, bachelor’s degree, more than bachelor’s degree master’s and PhD and unknown education status	Decedent’s educational attainment: what was highest educational diploma/degree achieved by the decedent?	
	Race/ethnicity	Non-Hispanic white, black, or African American, Hispanic/Latino, and other races	Is the decedent white? black? etc.	
County Health Rankings and Roadmaps	Adult smoking	Percentage	Percentage of adults that reported currently smoking (BRFSS)	
	Adult obesity	Percentage	Percentage of adults that report BMI >= 30 (CDC Diabetes Interactive Atlas)	
	Physical inactivity	Percentage	Percentage of adults that report no leisure-time physical activity (CDC Diabetes Interactive Atlas)	
	Excessive drinking	Percentage	Percentage of adults reporting binge or heavy drinking (BRFSS)	
	Unemployment	Percentage	Percentage of population ages 16+ unemployed and looking for work	
	Frequent mental distress	Percentage	Percentage of adults that reported having frequent mental distress (BRFSS)	
	Food insecurity	Percentage	Percentage of adults that reported being food insecure (Map the Meal Gap)	
	Insufficient sleep	Percentage	Percentage of adults that reported having insufficient sleep (BRFSS)	
	Children living in single-parent homes	Percentage	Percentage of children that live in a household headed by single parent (ACS)	

Computation of age-adjusted suicide mortality risks and cartographic displays

Suicide mortality risks were calculated and directly age-standardized to the 2010 US population in STATA version 8 using the “dstdize” command (Li et al., 2019; Mehmetoglu & Jakobsen, 2022). Using first-order queen contiguity spatial weights, spatial empirical Bayesian (SEB) smoothed risks were computed in GeoDa version 1.22.0.2 (Anselin, 2024). Risk smoothing was necessary to account for spatial autocorrelation and variance instability resulting from low population areas (Saman et al., 2012). Raw (or unsmoothed) age-adjusted suicide mortality risks and SEB risks were displayed as choropleth maps for each time period using Jenks’ classification scheme (Brewer & Pickle, 2002). All map visualizations were created using ArcGISPro version 3.2.0 (Environmental Systems Research Institute, 2024).

Investigation of high-risk clusters

To identify counties with significantly high suicide mortality risks, Tango’s Flexible Spatial Scan Statistics (FSSS) was used to detect both circular and irregegular shaped clusters using FleXScan version 3.1.2 (Takahashi, Yokoyama & Tango, 2010). The FSSS works by imposing a large number of overlapping, flexibly shaped windows of varying sizes over the study area to identify both circular and irregular shaped clusters (Tango & Takahashi, 2005). For this study, the specifications of Tango’s FSSS analysis were as follows: (i) Poisson probability model; (ii) restricted log likelihood ratio test (ensures that counties with low risk are not included in high-risk clusters); (iii) default alpha value of 0.2 (recommended by Tango & Takahashi (2005) and Tango & Takahashi (2012)); (iv) maximum scanning window size of 34 counties which is approximately half of the counties in the study area (adopted from Odoi (2020)); and (v) 999 Monte Carlo replications for statistical inference (Tango & Takahashi, 2005; Tango & Takahashi, 2012; Tango, 2021).

Investigation of predictors of county-level suicide mortality risk

Data for the most current available time period (2017–2019) were used to investigate predictors of county-level suicide mortality risks. Potential predictor variables used in the analysis were from 2018. Potential predictor variables considered in the investigation were selected based on a conceptual model (Fig. 2) (Posey, 2009; Kaplan et al., 2014; Blanchflower & Oswald, 2020; Kearns et al., 2020; Khader et al., 2020; Amiri, 2022; Graham & Ciciurkaite, 2023; Fabiano et al., 2024; Kareem et al., 2024; Blais et al., 2025). The conceptual model includes social determinants, behavioral risks factors, and mental health stressors that have been shown in past studies to be associated with suicide. Sociodemographic variables such as gender, race, age, education, veteran status, and marital status were investigated as potential predictors of exposure to adverse life conditions that impact risk of suicide. Previous studies have shown that veteran status, often characterized by effects of deployment stress, was associated with physical, psychological, and psychosocial outcomes which significantly elevates suicide risk particularly when combined with binge drinking (Posey, 2009; Blais et al., 2025). It has been reported that veterans who engaged in binge drinking were 33% more likely than non-veterans to report any suicide risk, and 72% more likely to plan a suicide without an attempt (Blais et al., 2025). This factor was also highlighted by Kaplan et al. (2014) who found that suicide decedents had significantly higher odds of acute alcohol use and intoxication prior to death. The sociodemographic factors included in the conceptual model have also been shown to be associated with unemployment, food insecurity, and single parenthood. Amiri (2022) reported that unemployment was strongly associated with suicide ideation, attempts, and mortality. Graham & Ciciurkaite (2023) showed that food insecurity increases suicide ideation among young adults, largely through elevated stress and social isolation. Similarly, Kareem et al. (2024) showed that single parenthood is linked to higher risk of depression and suicidality, emphasizing the emotional and physical toll of solo caregiving. Insufficient sleep, onset-insomnia, and sleep-related distress have been reported to be significant predictors of suicide ideation and mortality in adolescent and college student populations (Kearns et al., 2020; Khader et al., 2020). Additionally, a recent study showed that frequent mental distress has increased over time, particularly among economically disadvantaged populations (Blanchflower & Oswald, 2020), reinforcing its role as an important factor in suicide risk. Moreover, factors such as smoking, physical inactivity, obesity, and excessive drinking, often co-occur with frequent mental distress and may impact suicide risk (Fabiano et al., 2024). Thus, the conceptual model presented in Fig. 2 helped ensure that variable selection and model development were guided by evidence from existing scientific literature (Dohoo, Martin & Stryhn, 2012).

Figure 2 Conceptual model showing predictors of suicide.

To reduce the likelihood of multicollinearity, pairwise correlations of all potential predictors were assessed using two-way Spearman rank correlation analyses. Only one of a pair of the highly correlated variables (rs>—0.7—) was considered for further analysis. The choice of which of a pair of highly correlated potential predictors was retained for further analysis was based on literature review (biological plausibility) and statistical considerations. Potential predictors were further assessed for bivariate (unadjusted) associations with the outcome (county-level suicide mortality risk) using univariable negative binomial regression models with the number of suicide deaths as the outcome and the log of the total population per county as the offset. Negative binomial model was chosen because of presence of overdispersion in the data. Variables that were significantly associated with suicide mortality risk based on a liberal critical p ≤ 0.2 in the univariable analysis were included in subsequent multivariable modeling process (Dohoo, Martin & Stryhn, 2012). The assumptions of the negative binomial model include: linear relationship between the dependent and predictor variables, outcome is a count, no multicollinearity, presence of overdispersion (variance of the dependent variable is larger than its mean). This is a key feature that distinguishes a negative binomial model from a Poisson model. The form of the negative binomial model is: EY=nλorEYn=λ

where, E(Y) is the expected number of cases of suicide deaths, n is the measure of exposure, and λ is a function of the predictors.

The usual form of λ is derived from a linear equation on a natural log scale: 1nλ=β0+β1X1+β2X2+β3X3+…+βkXk.

This can also be expressed as: λ=eβ0+β1X1+β2X2+β3X3+…+βkXk

where, β0 is the intercept term representing the log count when all predictors are zero, β1…βk are the regression coefficients, and X1…Xk are the predictors.

The mean and variance are represented as follows: Mean:EY=μ

Variance:VarY=μ+αμ2

where, α is the dispersion parameter. If α = 0 it means the variance is equal to the mean and the model is then a Poisson model.

A manual backward elimination procedure was used to identify the most parsimonious multivariable negative binomial model using a critical p ≤ 0.05. Confounding was assessed by comparing the regression coefficients of variables in the model before and after the removal of a suspected confounder. A 20% or more change in coefficients of any variables in the model indicated confounding. Confounding variables were retained in the model regardless of their statistical significance. Biologically plausible two-way interaction terms of the variables in the final main effects model were also assessed and statistically significant ones retained in the final model. The coefficients from the final multivariable model were exponentiated and interpreted as risk ratios.

The goodness-of-fit of the final model was assessed using the chi-square goodness-of-fit test computed using both Pearson and the deviance residuals. Statistical analyses were performed using R statistical software version 4.3.2 (R Core Team, 2024), implemented in RStudio (R Studio Team, 2024). The geographical distributions of significant predictors were presented as choropleth maps created in ArcGISPro version 3.2.0 (Environmental Systems Research Institute, 2024).

Results

Geographical distribution

There were slight increases in the overall state age-adjusted suicide mortality risks from 22.6 deaths per 100,000 persons during the time periods 2011–2013, to 23.2 during 2014–2016, and 24.3 during 2017–2019 time periods. Although the spatial patterns of the SEB smoothed suicide mortality risks mirrored those of the raw risks, the spatial patterns of the SEB risks were more apparent/identifiable (Figs. 3 and 4). It is evident that the highest suicide mortality risks were consistently observed in the northwest, northeast, southwest, and some parts of the central regions of Florida during all three time periods (Figs. 3 and 4). Based on the SEB suicide mortality risk maps, there was evidence that the number of counties with high risks were increasing over time (Fig. 4).

Suicide clusters

Consistent with the spatial patterns of both the raw (unsmoothed) and SEB smoothed suicide mortality risks, clusters of high suicide mortality risks were identified consistently in the northwest and central regions (Fig. 5 and Table 2). Significant high suicide mortality risk clusters identified during the time periods 2011–2013 and 2014–2016 were in the northwest, central and southwest regions, whereas the high-risk clusters during 2017–2019 were in the eastern side of the northeast, and some parts of central, southwest and south regions. Seven counties were persistently in high-risk suicide mortality clusters throughout the study period. These counties were Volusia and Brevard counties in the eastern region; Citrus, Hernando, Pasco, and Pinellas counties in the southwest region; and Marion County in the central region of the state (Figs. 1 and 5).

Figure 3 Age-adjusted unsmoothed (raw) suicide mortality risks in Florida, 2011–2019.

The cartographic boundaries were obtained from the United States Census Bureau TIGER Geodatabase. Data source: Florida Department of Health. Map created using ArcGISPro.

Figure 4 Spatial empirical Bayes (SEB) smoothed suicide mortality risks in Florida, 2011–2019.

The cartographic boundary files were obtained from the United States Census Bureau TIGER Geodatabase. Data source: Florida Department of Health. Map created using ArcGISPro.

Figure 5 Geographic distribution of purely spatial clusters of high suicide mortality risks in Florida, 2011–2019.

The cartographic boundary files were obtained from US Census Bureau TIGER Geodatabase. Data source: Florida Department of Health. Map created using ArcGISPro.

Table 2 Purely spatial clusters of high suicide mortality risks in Florida, 2011–2019.

Time period	Cluster	Observed cases	Expected cases	Relative risk	p-value	
2011–2013	1	948	702	1.35	0.001	
2	182	132	1.38	0.022	
2014–2016	1	1,142	850	1.34	0.001	
	2	476	381	1.25	0.001	
3	276	207	1.33	0.004	
2017–2019	1	1,650	1,262	1.31	0.001	
2	217	163	1.33	0.024	

Predictors of geographic disparities in suicide mortality risks

A total of 16 potential predictors had significant univariable associations with suicide mortality risk based on a relaxed critical p = 0.2 (Table 3). Based on the final multivariable negative binomial model (Table 4), suicide mortality risks tended to be high in counties with high percentages of the population that: were aged 45–64 years (RR = 1.010; p = 0.017), were aged ≥ 65 years (RR = 1.008; p = 0.016), drank alcohol excessively (RR = 1.064; p < 0.001), had frequent mental distress (RR = 1.039; p < 0.001), and were veterans (RR = 1.022; p = 0.005). Although not statistically significant insufficient sleep (p = 0.264) was included in the model because it was an important confounder of age 65 years and above. The significant (p = 0.010) interaction between excessive drinking and insufficient sleep implies that the effect of the percentage of population involved in excessive drinking on county risk of suicide deaths depends on the percentage of population in the county that experience insufficient sleep (Table 4). The risk ratios seem relatively small because they represent the effect of 1% increase in each of the variables. Higher increases in the percentages of the significant predictors would result in much higher effects. For instance, a 10% increase in population reporting frequent mental distress would result in a 2.34 times higher risk of suicide deaths (e(0.085∗10) = 2.34) at the county level when all other variables are kept constant. There was no evidence of either multicollinearity (variance inflation factor (VIF) were all below 1.9) or lack of fit (Pearson Chi-Square tests, p = 0.26; deviance chi-square test, p = 0.33).

Table 3 Univariable associations between suicide mortality risks and potential predictors in Florida, 2017–2019.

Predictor (Percentage of population)	Median	IQR a	Coefficient (95% CI b )	RR c (95% CI b )	p-value	
Male	77.8	75.8–82.3	0.007 (−0.002–0.016)	1.007 (0.998–1.016)	0.136*	
Female	22.2	17.5–24.1	0.001 (−0.009–0.012)	1.001 (0.991–1.012)	0.824	
18 to 44 years	31.9	25.7–40.0	−0.007 (−0.014–0.000)	0.993 (0.986–1.0)	0.039*	
45 to 64 years	38.3	34.3–42.2	0.010 (0.000–0.02)	1.010 (1.001–1.020)	0.039*	
65+ years	25.2	20.0–32.4	0.008 (0.000–0.015)	1.008 (1.001–1.015)	0.038*	
Less than 18 years	1.9	0.0–3.1	−0.020 (0.048–0.007)	0.980 (0.953–1.007)	0.154*	
Never married	26.5	22.8–33.3	−0.004 (−0.012–0.005)	0.996 (0.998–1.005)	0.359	
Widowed	8.4	5.4–11.0	0.018 (0.002–0.033)	1.018 (1.002–1.034)	0.026*	
Divorced	23.2	20.2–27.4	0.006 (−0.005–0.017)	1.006 (0.995–1.017)	0.277	
Married	36.8	32.0–42.2	0.002 (−0.007–0.010)	1.002 (0.994–1.010)	0.625	
Less than high school education	13.6	10.2–19.3	0.003 (−0.007–0.014)	1.003 (0.993–1.014)	0.527	
High school education	39.0	34.2–47.2	0.005 (−0.003–0.013)	1.005 (0.997–1.013)	0.206	
Some college	15.8	11.9–19.6	0.004 (−0.007–0.016)	1.004 (0.993, 1.0159)	0.445	
Associate degree	8.3	6.1–10	0.010 (−0.012–0.031)	1.010 (0.988–1.032)	0.372	
Bachelor’s degree	11.8	7.6–16.1	−0.010 (−0.025–0.003)	0.989 (0.975–1.003)	0.109*	
More than bachelor’s degree	5.8	0.0–8.5	−0.003 (−0.020–0.014)	0.997 (0.980–1.014)	0.729	
Non-Hispanic white	95.2	90.9–100	0.025 (0.012–0.038)	1.025 (1.012–1.039)	0.000*	
Black or African American	2.1	0.0–6.1	−0.030 (−0.049–0.011)	0.970 (0.952–0.989)	0.002*	
Hispanic/Latino	5.6	0.0–10.9	−0.013 (−0.0186–0.006)	0.988 (0.982–0.994)	<0.000*	
Smokers	21.5	19.2–25.6	0.021 (0.000–0.042)	1.021 (1.000–1.042)	0.035	
Obese	31.8	27.3–36.2	0.005 (−0.010–0.019)	1.005 (0.990–1.019)	0.509	
Excessive drinking	19.4	18.1–21.5	0.029 (−0.003–0.060)	1.029 (0.997–1.062)	0.072*	
Unemployment	3.4	3.1–3.8	0.103 (−0.011–0.218)	1.109 (0.989–1.244)	0.079*	
Frequent mental distress	15.7	14.7–17.7	0.043 (−0.003–0.089)	1.044 (0.998–1.092)	0.053*	
Food insecure	13.7	12.2–15.4	0.014 (−0.023–0.051)	1.014 (0.978–1.052)	0.437	
Insufficient sleep	39.8	38.5–41.0	0.029 (−0.005–0.063)	1.030 (9.950–1.065)	0.086*	
Children in single parenthood	36.4	33.4–39.8	0.003 (−0.010–0.015)	1.003 (0.990–1.015)	0.660	
Veteran	10.8	8.9–12.6	0.041 (0.020–0.060)	1.042 (1.021–1.063)	<0.000*	
Notes.

a Interquartile range.

b 95% Confidence interval.

c Risk ratio (exponentiated coefficients) and corresponding 95% confidence interval.

* Significant variables alpha of <0.20.

Table 4 Significant predictors of geographic disparities of suicide mortality risks in Florida, 2017–2019.

Predictor variables	Estimate (95% CIa)	RRb (95% CIa)	P-Value	VIF c	
Age 45 to 64 years	0.009 (0.002, 0.017)	1.010 (1.002, 1.018)	0.016	1.16	
Age 65 and above	0.006 (0.002, 0.012)	1.006 (1.001, 1.013)	0.034	1.21	
Excessive Drinking	0.062 (0.030, 0.094)	1.064 (1.031, 1.098)	<0.001	2.25	
Insufficient Sleep	0.019 (0.0, 0.053)	1.019 (0.986, 1.054)	0.264	2.0	
Frequent mental distress	0.085 (0.038, 0.131)	1.089 (1.039, 1.140)	<0.001	1.96	
Veterans	0.022 (0.006, 0.038)	1.022 (1.007, 1.039)	0.005	1.22	
Excessive Drinking X Insufficient Sleep	0.015 (0.003, 0.026)	1.015 (1.003, 1.027)	0.010	1.57	
Notes.

a 95% Confidence Interval.

b Risk ratio.

c Variance inflation factor.

Counties in the northwest and northeast regions of Florida tended to have high percentages of the population: aged 45–64 years, with frequent mental distress, and who were veterans (Fig. 6). Counties with high percentages of population who drank excessively were in the northeast, southwest, south, and the eastern side of central and southeast regions of the state while those with a high percentage of populations aged 65 years and above tended to be in the northwest, central, and southwest regions of the state.

Figure 6 Geographic distribution of significant predictors of county-level suicide mortality risks in Florida, 2017–2019.

The cartographic boundary files were obtained from the publicly available United States Census Bureau TIGER Geodatabase. Data source for percentage of those aged 45 to 64 years, and above 65 years: Florida department of health. Data source for percentage of excessive drinking, and frequent mental distress: county health rankings and road maps. Data source for percentage of veterans: US Census Bureau, American Community Survey, Veteran Status S2101. Map created using ArcGISPro.

Discussion

This study investigated county-level geographic disparities between 2011 and 2019 and predictors of suicide mortality risks in Florida between 2017 and 2019. Results show evidence of geographic disparities in suicide mortality risks in the state. A number of predictors of the identified disparities were identified. The findings provide useful information to guide needs-based resource allocation to reduce disparities in suicide mortalities (Saman et al., 2012; O’Rourke, Jamil & Siddiqui, 2018).

Similar to findings from other studies investigating disparities in suicide mortalities in the US (Saman et al., 2012; Johnson et al., 2017; Garnett & Curtin, 2023) the present study found disparities in the burden of suicides in Florida, with the northwest and central regions having excessively high suicide mortality risks. It is noteworthy that high-risk suicide clusters tended to occur in rural counties. These findings are consistent with those of some previous studies (Saman et al., 2012; Fontanella et al., 2015; Lee et al., 2023). Some of the counties included in the high-risk suicide clusters (e.g., Volusia, Marion, Citrus, Brevard, Hernando, Pasco, and Pinellas) are known to have several risk factors for suicide (Donley, Fernandez-Reiss & Austin, 2023). For instance, these counties have high rates of mental illnesses such as depressive disorders, as well as high rates of hospitalizations due to drug and alcohol induced mental disorders (Facioli et al., 2015). Poor access to healthcare facilities, shortage of healthcare professionals, and overburdened healthcare systems may also contribute to the high suicide risks observed in these areas (Donley, Fernandez-Reiss & Austin, 2023).

The observed association between suicide mortality risk and age is consistent with findings from another study which reported that risk of suicide tended to increase with age (Crestani et al., 2019). A different study in Florida reported that areas with a high elderly population were likely to be in high suicide mortality risk clusters (Johnson et al., 2017). This finding is consistent with those of the current study. It is noteworthy that suicide is the 8th leading cause of death among the 55–64 year-olds and the 15th leading cause of death among those aged 65 years and above in Florida (Hansen & Gryglewicz, 2016). Unfortunately, a study by Crestani et al. (2019) reported that the problem of suicides among the elderly population is not given as much attention as among younger individuals. In the current study, clear patterns of high proportion of elderly populations (aged ≥ 65 years), were observed in the northwest and southwest regions. This might explain the high-risk clusters of suicide mortality risks observed in those areas. However, to avoid ecological fallacy, direct inference can only be made at the individual level.

The association between county-level suicide risk and the percentage of population with frequent mental distress observed in the current study may be due to the relatively high level of mental illness in the state. For instance, the Florida Health Justice Project reported that approximately one-fifth (17.5%) of Floridians aged 18 years and older have a mental illness and over half a million adults in the state experience serious mental illness (Yager, 2020). This is exacerbated by the fact that 60% of adults with mental illness in Florida do not receive treatment (Welty et al., 2019). Studies have also reported that untreated, under-treated, or unidentified mental illness/distress were risk factors of suicide across all populations (Rossen et al., 2018; Perry et al., 2022). It is worth noting that there is a significant shortage of mental health services in Florida which is ranked 40th out of the 50 US states in terms of access to mental health services and 49th in total spending on mental illness (Yager, 2020).

The finding of an association between county-level risk of suicide and the percentage of veterans in the county is consistent with reports from other studies that have reported high risk of suicides among veterans (Carney, 2014; VA Office of Mental Health and Suicide Prevention, 2023; Finnegan, Salem & Ainsworth-Moore, 2024). According to the US Department of Veterans Affairs (VA), 6,392 veterans died by suicide in 2021, representing a 11.6% increase from the previous year. In the same year, suicide was the 13th leading cause of death among veterans and the 2nd leading cause of death among those under 45 years of age (Centers for Disease Control and Prevention, 2023; VA Office of Mental Health and Suicide Prevention, 2023). Most (70%) of the veterans who committed suicide were 50 years or older, which was twice the suicide rate for the same age group among civilians (Florida Department of Children and Families, 2024). It is worth noting that the findings from this study should be interpreted with caution to avoid ecological fallacy since inferences should not be made at the individual level based on these county-level analyses and results.

The association between high rates of excessive drinking and high county-level suicide mortality risk is consistent with findings from previous studies which reported that the co-occurrence of alcohol use disorders and depression heightened suicide risks (Blow, Brockmann & Barry, 2004; Xuan et al., 2016). There is evidence that alcohol abuse may lead to suicidal tendencies as a result of impaired judgment (Pompili et al., 2010). In addition, the significant interaction observed between excessive drinking and insufficient sleep suggests the combined effects of excessive drinking and insufficient sleep may contribute to a higher suicide risk. These results emphasize the need for integrated public health strategies that prioritize sleep hygiene and substance use prevention, particularly in populations at higher risk for suicide. Future research should further explore these interactions at the individual level to better inform suicide prevention efforts.

Strengths and limitations

Use of SEB age-adjusted mortality risks ensure adjustment for both spatial autocorrelation and risk instability due to high variances in low population counties. The implementation of a FSSS with a restricted likelihood ratio enables the identification of both circular and non-circular clusters. Moreover, employing a restricted likelihood ratio test in place of the original log likelihood ratio minimizes false positives. The limitation of the study is that it was conducted at the county-level and hence patterns observed at this geographic scale may be different from those at lower geographic scales. Unfortunately, due to small numbers, the study could not be conducted at levels lower than the county spatial scale.

Conclusions

There are persistent geographical disparities in suicide burden across Florida, with the northwest, northeast, southwest, and some parts of the central regions showing significantly higher mortality risks than the rest of the state. Suicide prevention efforts should be targeted at the high-risk counties. A number of county-level predictors of suicide mortality were identified, including high percentages of individuals aged 45–64 years and ≥65 years, excessive drinking, frequent mental distress, insufficient sleep, and veterans. Efforts to address the problem should focus not only on high-risk areas but also the identified predictors of the identified geographic disparities.

Supplemental Information

Supplemental Information 1 Raw Study Data and Codebook

All variables used in the analyses.

The authors are grateful to the Florida Department of Health for providing the data used in this study.

Additional Information and Declarations

Competing Interests

Author Contributions

Human Ethics

Ethics

Data Availability

Agricola Odoi is an Academic Editor for PeerJ.

Howard Onyuth conceived and designed the experiments, performed the experiments, analyzed the data, prepared figures and/or tables, authored or reviewed drafts of the article, and approved the final draft.

Corey Day analyzed the data, authored or reviewed drafts of the article, and approved the final draft.

Nirmalendu Deb Nath analyzed the data, authored or reviewed drafts of the article, and approved the final draft.

Agricola Odoi conceived and designed the experiments, performed the experiments, analyzed the data, prepared figures and/or tables, authored or reviewed drafts of the article, and approved the final draft.

The following information was supplied relating to ethical approvals (i.e., approving body and any reference numbers):

The University of Tennessee Institutional Review Board approved this study (IRB-24-083540-XM).

This study was reviewed by the University of Tennessee Institutional Review Board (IRB) which determined that it was not human subjects research (IRB Number: UTK IRB-24-083540-XM) because the secondary data used in the study did not contain information on living individuals. Therefore, the board determined that IRB oversight was not required.

The following information was supplied regarding data availability:

The raw data is available in the Supplementary File.

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
