# Peer review of "Geographic disparities and predictors of suicide mortality risk in Florida: spatial scan statistics and negative binomial modeling"

_PeerJ, doi:10.7717/peerj.20075_

## Round 0.1 · original submission · Major Revisions

· Academic Editor

Major Revisions

Please address the comments of the reviewers. Pay particular concern to the comments from R1 and R2

Reviewer 1 ·

Basic reporting

The paper is very well structured and written which investigated the spatial disparities of suicide and potential factors that explain the patterns in Florida.

Minor points:
In Introduction, line 115, suggesting the authors to change "substance abuse" to "substance use" to avoid negative judgements.
In Materials & Methods, I'm not sure if the "Ethics Review" section is required by the journal, but that paragraph seems very repetitive.
In Materials & Methods, the sentence from lines 137-139 is broken.

Experimental design

In Figure 1, it is unclear how counties were classified into mostly rural, mostly urban, and completely rural. The information was not provided in the listed reference.

The entire section of "Investigation of high-risk clusters" is confusing to me.

lines 174-175: "Tango's Flexible Spatial Scan Statistics (FSSS) was used to identify both circular and irregularly shaped clusters using FleXScan version 3.1.2." -- why was this step needed and what were the results?
In the next sentence, they mentioned the use of a Poisson model. Why didn't they use negative binomial (as in the risk factors sections) models with an offset, since over-dispersion of the outcome should be an issue here too and one should account for the total number of population in each county. Also, what are the independent variables in this model? It seems to be the clusters, but please make it clear.
It's also unclear why the alpha level was set to .2 and cluster size was set to be less than 34 counties (that's still more than half of the total number of counties of Florida).

In the "Investigation of predictors of County-level Suicide Mortality Risk" section, please provide a reference for Figure 2. Figure 2 is also hard to read. I'd suggest uploading a higher resolution figure. According to Figure 2, some factors might be interacting with each other. I don't think the interaction effects between factors were considered using the current variable selection schema. Maybe this point should be discussed in the limitation.

Similarly, in lines 202-203, the authors mentioned biologically plausible interactions were assessed. But no interactions were reported in Table 4. Which interactions were assessed? And were none of them significant?

I'd recommend the authors submitting the manual variable selection process as supplementals. It seems like some predictions with stronger independent associations were not selected (eg, latinx had p<0.0001 in Table 3), regardless "Frequent Mental Distress" was selected in the final model with both independent and joint p-values >0.05. I even start wondering if the authors picked p<0.2 as the criteria just so some of their hypothesized predictors could be selected.

Death data are from 2011-2019. But when were the covariates in Table 1 collected? How were the death data aggregated in the models need to be mentioned.

Please round up the coefficients in Table 3. Also, if you were fitting negative binomial models, shouldn’t the coefficients here log(RR)? Then recommend exponentiate them. Same for table 4.

Validity of the findings

I don't think the death data were provided in the uploaded raw data.

·

Basic reporting

Title- I suggest including negative binomial regression at the title.

Introduction
Your introduction needs more detail. I suggest that you improve the writing of the sentences at line 87 to provide more clearer sentences, by adding ‘behavior’ instead of suicide to suicide behavior. Do the same thing for every word suicide which is to change suicide to suicide behavior.

Line 90- I suggest you improve the sentences from one to 2 sentences. Approximately 800,000 people worldwide die of suicide behaviour annually, meaning that on average, one person dies from suicide every 40 seconds (Martinez-Ales et al., 90 2020) is one sentence. Suicide accounts for a higher death rate compared to fatalities from malaria, HIV/AIDS, breast cancer, war, or homicide (Hisham et al., 2023) become one sentences. The English language should be improved to ensure that an international audience can clearly understand your text.– the current phrasing makes comprehension difficult. I suggest you have a colleague who is proficient in English and familiar with the subject matter review your manuscript, or contact a professional editing service in this sentences.

Line 108- I suggest that you improve the writing of the sentences at line 108 to provide more clearer sentences, by adding ‘reported’ before the word every 2 hours.

Line 111-I suggest that you improve the writing of the sentences at line 111 to provide more clearer sentences, by adding ‘in the age of ….’ Before 500,000 adults.

Line 116-I suggest that you improve the writing of the sentences at line 116 to provide more clearer sentences, by adding ‘behaviour’ before (Brenes et al, 2023).

Line 118- Your research gap needs more detail. I suggest that you improve the description at lines 118 to provide more justification for your study (specifically, you should expand upon the knowledge gap being filled).

Ethics review
Remove ‘Informed consent was not possible or needed since it was determined that it was not a human subjects study and study data only contained death records and no information on living individuals.’ from the paragraph. Meanwhile, the remaining sentence ‘This study was reviewed by the University of Tennessee, Knoxville Institutional Review Board (IRB) which determined that it was not human subjects research (IRB Number: UTK IRB-24-083540-XM) because all the information obtained from the secondary dataset that used in the study contains death data with no information on living individuals. Therefore, the review board determined that IRB oversight was not required’ need for professional editing service in this sentences.


Data sources
Line 143- I suggest improving the sentences to ‘The secondary data used in the study were obtained from the Florida Department of Health (source). Then, the data were split according to county spatial and divided into 3-year periods: 2011-2013, 2014-2016, and 2017-2019’. It would be more appropriate to use the term "geographical " instead of "spatial " to maintain consistency in terminology throughout the text.

Line 145- I suggest improving the sentence to ‘Death data records of all deceased Florida residents were needed in the study’.

Line 146- I suggest improving the sentences by adding ‘The cause’ to line 146. The cause of death from suicide was identified using the International Classification of Diseases, 10th revision (ICD-10) External Cause of Death Codes (X60-X84, Y87.0), which met the Centers for Disease Control and Prevention (CDC) National Violent Death Reporting System definition of suicide (Saman et al., 2012).

Line 149-152-I suggest adding citations in this line regarding the survey.

Line 164-166-I suggest adding ‘and’. Additionally, smoothed risks were calculated using spatial empirical Bayesian (SEB) rate smoothing in GeoDa version 1.22.0.2 (Anselin, 2024), and using first-order queen contiguity spatial weights.

Line 178, 193,197- Why use alpha 0.2 instead of 0f 0.05? Please justify.

Line 217- I suggest including the classification of these regions (such as northwest, central regions of Florida, etc) in the study area section. In the study area section, instead of discussing the ethnicity of the country, why not explaining the percentage of the population in each region?

Line 235- I suggest to improve the style of writing such as follows; Based on the final multivariable negative binomial model, the risk of suicide mortality risks tended to be high in counties with high percentages of population: aged 45-64 years (IQR=…..;p-value=…..) , aged 65 years (IQR=…..;p-value=…..), who drank alcohol excessively (IQR=…..;p-value=…..), with frequent mental distress (IQR=…..;p-value=…..), and who were veterans (IQR=…..;p-value=…..) (Table 4).

Line 257- I suggest to improve the style of writing such as follows; Results from this study also reveal that high-risk suicide clusters tended to occur in rural counties. These findings are consistent with those of some previous studies (Saman et al., 2012; Fontanella et al., 2015; Lee et al., 2023), but contrary to those of others (Casant & Helbich, 2022; Satherley et al., 2022).

Line 272- I suggest to improve the style of writing such as follows; A different study in Florida reported that areas with a high elderly populations were likely to be in high suicide mortality risk clusters (Johnson et al., 2017). I suggest to improve the style of writing such as follows; The finding consistent with those of the current study. It is noteworthy that suicide is the 8th leading cause of death among the 55-64 year olds and the 15th leading cause of death among those aged 65 and above in Florida (Hansen & Gryglewicz, 2016).

line 278- I suggest improving the style of writing such as follows; In the current study, clear patterns of the high proportion of elderly populations (aged 65 years), were observed in the northwestern region. This might explain, the high-risk clusters of suicide mortality risks observed in those areas in the elderly population. However, to avoid ecological fallacy, direct inference can only be made at the individual following individual level study that would assess if the high suicide mortalities in these areas are, in fact, due to seniors taking their own lives.

Overall
Need proofread. The English language should be improved to ensure that an international audience can clearly understand your text.
Must use consistent use of word such as spatial change to geographic.
Need to reduce the references because too many.
Need to reduce number of pages. The total current page number is 44. I suggest to reduce to 15 pages.

Experimental design

Typically, a significance level (alpha) of 0.05 is used rather than 0.2. I recommend that the authors adjust the threshold to 0.05 and re-evaluate the significance of the predictors, ensuring that only variables with p-values < 0.05 are considered significant.

Additionally, I suggest including the negative binomial regression equation in the manuscript. It would also be beneficial to describe the assumptions required to run this type of regression analysis, providing clarity and justification for its application in the study.

Rewrite the methodology section in line 181-209, so that it shows the flow of statistical data analysis that has been done to run the output.

Validity of the findings

Can cut the leng

Additional comments

roofreading and Language Improvement:
The manuscript requires thorough proofreading to enhance the quality of English. Improving the clarity and grammar of the text is essential to ensure that an international audience can easily understand your work. I suggest consulting a professional editing service or a colleague proficient in English and familiar with the subject matter.

Consistency in Terminology:
There is an inconsistency in the use of terminology, such as "spatial" and "geographic." I recommend standardizing the terminology throughout the manuscript. For instance, replace "spatial" with "geographic" for consistency and accuracy.

Reduction in References:
The manuscript currently cites an excessive number of references. To improve focus and readability, I suggest reducing the number of references by retaining only the most relevant and high-impact citations.

Page Length:
At present, the manuscript is 44 pages long. This is significantly lengthy and may hinder readability. I recommend reducing the manuscript to approximately 15 pages, focusing on the core findings and key discussions to enhance its impact and clarity.

·

Basic reporting

The article is written in clear and crisp language. It provides a concise introduction with a relevant research quoted wherever required. With the use of appropriate methods, results were presented in perscribed style, the authors reach to a conclusion. References are presented as per the format.
The article may be accepted as it is.

Experimental design

The article utilizes appropriate methods to address the problem of research. With a specific period of investigation, the author justifies the extent and limit of the study.

Validity of the findings

With well stated conclusions the study guides the researchers to take the research further, and encourages its replication in the other areas of world.

Additional comments

Congratulations for contributing a research that will help understand the issue of Suicide better. The study is well thought of, well designed and executed well.

---

## Round 0.2 · Major Revisions

· Academic Editor

Major Revisions

Reviewer 1 ·

Basic reporting

I appreciate the authors’ effort in addressing my comments during the revision. However, I have a few minor comments arising from the revised manuscript.

The authors added “negative binomial” to the title (as requested by the other reviewer)— however, to me, this study wasn’t completely based on negative binomial models. The first half examined the geographic high-risk clusters, and only the second half used negative binomial models to identify predictors. Including “a negative binomial modeling approach” in the title is misleading.

What does “Negative statistical analyses” mean (line 356, page 10)?

Experimental design

The responses to my questions related to FSSS and Poisson model made me worry if the authors truly understand the methods. Was a Poisson model used by FSSS to identify clusters? If this is the case, the authors should make it clear in text. Otherwise, readers might think the Poisson model was another step they further conducted after the FSSS. If the alpha=0.2 and cluster size of 34 were recommended by certain papers, they should mention it in the text, not just providing citations.

I appreciate the authors highlighting the suicide risk predictors were investigated using the 2017-2019 time period. However, the covariates were from the 2018 data were not mentioned in the referenced section. Similarly, the fact that predictors were only evaluated in the 2017-2019 data should be reflected in the abstract and the discussion section (first sentence, for example).

If the authors came up with Figure 2, then what’s their justification for the connections and directions presented in the figure? The figure was included to justify the potential predictors they considered, but the figure was not backed up by any references. In another response, the author mentioned that they explained all the two-way interactions between the main effects in the final main-effect model — then their potential modeling schema was not based on Figure 2, as stated by them.

Validity of the findings

I appreciate the authors for providing the data, but if you provide the data, then a data codebook should be included as well to explain all the variables. What columns in the “study_data.xlsx” reflect the death data? With guessing, I could identify most of the independent variables, but not death data (outcome).

·

Basic reporting

Overall Evaluation:
The manuscript has shown notable improvement. The text is now well-structured, clearly written, and technically sound in most areas. The author has made commendable progress in refining the manuscript's clarity and coherence.

However, there are still a few technical issues that require attention:

Line 117: The uppercase ‘L’ should be replaced with a lowercase ‘l’ to maintain consistency with standard notation.

Line 225: A citation is needed to support the statement involving p < 0.2. Please provide a reference or rationale to justify the statistical threshold used.

Aside from these minor corrections, the manuscript is in good shape.

Congratulations to the author on the improvements made thus far.

Experimental design

no comment

Validity of the findings

no comment

Additional comments

no comment

---

## Round 0.3 · Minor Revisions

· Academic Editor

Minor Revisions

Please address the remaining point from Reviewer 1.

Reviewer 1 ·

Basic reporting

I appreciate the authors for addressing all my comments. My only remaining comment is that, while references were added to explain Figure 2, no accompanying text explanation was included.

Experimental design

NA

Validity of the findings

NA

Additional comments

NA

---

## Round 0.4 · accepted · Accept

· Academic Editor

Accept

We're happy with the current version of your manuscript.

Reviewer 1 ·

Basic reporting

Thank you for addressing all my comments.

Experimental design

-

Validity of the findings

-